# Interactive Image Generation Using Scene Graphs

**Gaurav Mittal**[\*][†], **Shubham Agrawal**[\*], **Anuva Agarwal**[\*][†], **Sushant Mehta**[\*][‡] **& Tanya Marwah**[\*]
Carnegie Mellon University
Pittsburgh, PA 15213, USA
{gauravm, sagrawa1, anuvaa, sushantm, tmarwah}@andrew.cmu.edu

## Abstract

Recent years have witnessed some exciting developments in the domain of generating images from scene-based text descriptions. These approaches have primarily focused on generating images from a static text description and are limited to generating images in a single pass. They are unable to generate an image interactively based on an incrementally additive text description (something that is more intuitive and similar to the way we describe an image). We propose a method to generate an image incrementally based on a sequence of graphs of scene descriptions (scene-graphs). We propose a recurrent network architecture that preserves the image content generated in previous steps and modifies the cumulative image as per the newly provided scene information. Our model utilizes Graph Convolutional Networks (GCN) to cater to variable-sized scene graphs along with Generative Adversarial image translation networks to generate realistic multi-object images without needing any intermediate supervision during training. We experiment with Coco-Stuff dataset which has multi-object images along with annotations describing the visual scene and show that our model significantly outperforms other approaches on the same dataset in generating visually consistent images for incrementally growing scene graphs.

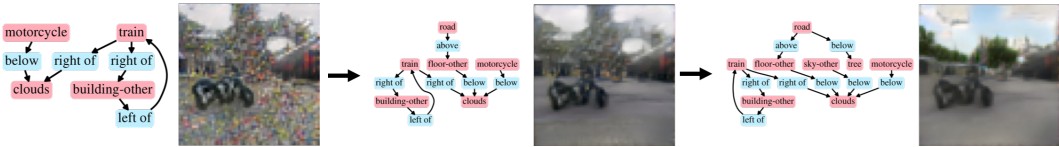

Figure 1: We propose an image generation framework capable of generating images from scene graphs, and then modify the image based on modifications to the scene graph, without losing previously generated image content.

## 1 Introduction

To truly understand the visual world, our models should be able to not only recognize images but also generate them. Generative Adversarial Networks, proposed by Goodfellow et al. (2014) have proven immensely useful in generating real world images. GANs are composed of a generator and a discriminator that are trained with competing goals. The generator is trained to generate samples towards the true data distribution to fool the discriminator, while the discriminator is optimized to distinguish between real samples from the true data distribution and fake samples produced by the generator. The next step in this area is to generate customized images and videos in response to the individual tastes of a user. A grounding of language semantics in the context of visual modality has wide-reaching impacts in the fields of Robotics, AI, Design and image retrieval. To this end, there has

---

[\*]Authors contributed equally.
[†]Currently working at Microsoft Research.
[‡]Currently working at Google.

been exciting recent progress on generating images from natural language descriptions. Conditioned on given text descriptions, conditional-GANs (Reed et al., 2016) are able to generate images that are highly related to the text meanings. Samples generated by existing text-to-image approaches can roughly reflect the meaning of the given descriptions, but they fail to contain necessary details and vivid object parts.

Leading methods for generating images from sentences struggle with complex sentences containing many objects. A recent development in this field has been to represent the information conveyed by a complex sentence more explicitly as a scene graph of objects and their relationships (Johnson et al., 2018). Scene graphs are a powerful structured representation for both images and language; they have been used for semantic image retrieval (Johnson et al., 2015) and for evaluating (Anderson et al., 2016) and improving (Liu et al., 2017) image captioning. In our work, we propose to leverage these scene graphs by incrementally expanding them into more complex structures and generating corresponding images. Most of the current approaches lack the ability to generate images incrementally in multiple steps while preserving the contents of the image generated so far. We overcome this shortcoming by conditioning the image generation process over the cumulative image generated over the previous steps and over the unseen parts of the scene graph. This allows our approach to generate high quality complex real-world scenes with several objects by distributing the image generation over multiple steps without losing the context. Recently, El-Nouby et al. (2018) proposed an approach for incremental image generation but their method is limited to synthetic images due to the need of supervision in the intermediate step. Our approach circumvents the need for intermediate supervision by enforcing perceptual regularization and is therefore compatible with training for even real world images (as we show later).

A visualization of our framework's outputs with a progressively growing scene graph can be seen in Figure 1. We can see how at each step new objects get inserted into the image generated so far without losing the context. To summarize, we make the following contributions,

- We present a framework to generate images from structured scene graphs that allows the images to be interactively modified, while preserving the context and contents of the image generated over previous steps.

- Our method does not need any kind of intermediate supervision and hence, is not limited to synthetic images (where you can manually generate ground truth intermediate images). It is therefore useful for generating real-world images (such as for MS-COCO) which, to the best of our knowledge, is the first attempt of its kind.

## 2 RELATED WORK

Generating images from text descriptions is of great interest, both from a computer vision perspective, and a broader artificial intelligence perspective. Since the advent of Generative Adversarial Networks (GANs), there have been many efforts in this direction (Huang et al., 2018). Ouyang et al. (2018) proposed a framework based on an LSTM and a conditional GAN to incrementally generate an image using a sentence. The words in the sentence were encoded using word2vec, and passed through an LSTM. A skip-though vector representing the semantic meaning of an entire sentence is used as the conditioning for the GAN. However, all of these works mostly focus on generating images with single objects (such as faces or flowers or birds). Even within these objects, the avenues of variance is quite limited. Generating more complex scenes with multiple objects and specific relationships between those objects is an even harder research problem.

Zhang et al. (2017a) proposed an architecture based on multiple GANs stacked together, generating images in a coarse-to-fine manner. They later also proposed arranging the generators in a tree-like structure for improved results (Zhang et al., 2017b). More recently, Hong et al. (2018) proposed an end-to-end pipeline for inferring scene structure and generating images based on text descriptions. A similar approach was taken by Tan et al. (2018), where they used attention-based object and attribute decoders to infer bounding box locations of objects in the scene. However, for images with several objects such as in COCO-Stuff, the captions are often not descriptive enough to capture all the objects. Furthermore, the captions don't describe the relations between the objects in the image effectively. AttnGANs also begin with a low-resolution image, and then improve it over multiple steps to come

up with a final image. However, there's no mechanism to capture consistency during incremental image generation. A more detailed failure case analysis is done in Section 4.

Most recently, Johnson et al. (2018) proposed to use scene-graphs as a convenient intermediate for image synthesis. Scene-graphs provide an efficient and interpretable representation of the objects in an image and their relationships. The input scene graph is processed with a graph convolution network which passes information along edges to compute embedding vectors for all objects. These vectors are used to predict bounding boxes and segmentation masks for all objects, which are combined to form a coarse scene layout. The layout is passed to a cascaded refinement network which generates an output image at increasing spatial scales. The model is trained adversarially against a pair of discriminator networks which ensure that output images look realistic. However, this model does not account for object saliency or temporal consistency in the generated images. It also fails for highly complex scene where there are several objects due to a single-pass image generation. A more detailed failure case analysis is done in Section 4.

Another recent work by El-Nouby et al. (2018) introduces a conditional text-to-image based generation approach to generate images iteratively, keeping track of ongoing context and history. Their method uses GRU to process text instructions into embeddings which go through conditioning augmentation and are fed into a GAN to generate contents on a canvas. They use a convex combination of feature maps to ensure consistency. Due to the availability of structured information in the form of scene graphs, we can instead filter out the latent embeddings of the already generated objects, thus allowing for better textural consistency. Moreover, their approach suffers from the drawback of needing supervision at every stage of generation for training and is therefore limited to only synthetic scenarios (such as CoDraw and i-CLEVR where intermediate images can be easily rendered). On the contrary, our approach employs perceptual similarity based regularization and effective use of graph-based embeddings to circumvent the need of ground truth for intermediate steps, making it compatible with even real-world images.

## 3 Method

As in most modern conditional image-generation formulations, we follow a generative-adversarial approach to image generation. Here the adversarial network penalizes the network based on how realistic the generated images are, as well as whether the required objects are present in them. Furthermore, a key part of the task is to preserve the relations between objects as specified in the scene graph in the generated image as well.

### 3.1 Image generation from scene graphs

For our baseline approach for generating images from scene graphs, we adopt the architecture proposed by Johnson et al. (2018). The architecture consists of 3 main modules, a Graph Convolution network (GCN), Layout Prediction Network (LN) and a Cascade Refinement Network (CRN), which we describe in more detail below. For brevity, we omit an exhaustive background description of these modules and request the reader to refer to the original paper for further details. Figure 2 provides an overview of the full model architecture.

**Graph Convolution Network.** The Graph Convolution Network (GCN) is composed of several graph convolution layers, and can operate natively on graphs. GCN takes an input graph and computes new vectors for each node and edge. Each graph convolution layer propagates information along edges of the graph. The same function is applied to all graph edges, which ensures that a single convolution layer can work with arbitrary shaped graphs.

**Layout Prediction Network.** The GCN outputs an embedding vector for each object. These object embedding vectors are used by the layout prediction network to compute a scene layout by predicting a segmentation mask and bounding box for each object. Mask regression network and a box regression network are used to predict a soft binary mask and a bounding box, respectively. The layout prediction network hence acts as an intermediary between the graph and image domains.

**Cascade Refinement Network.** Given a scene layout, the Cascade Refinement Network (CRN) is responsible for generating an image which respects the object positions in the scene layout. The CRN consists of a series of convolutional refinement modules. The spatial resolution doubles between

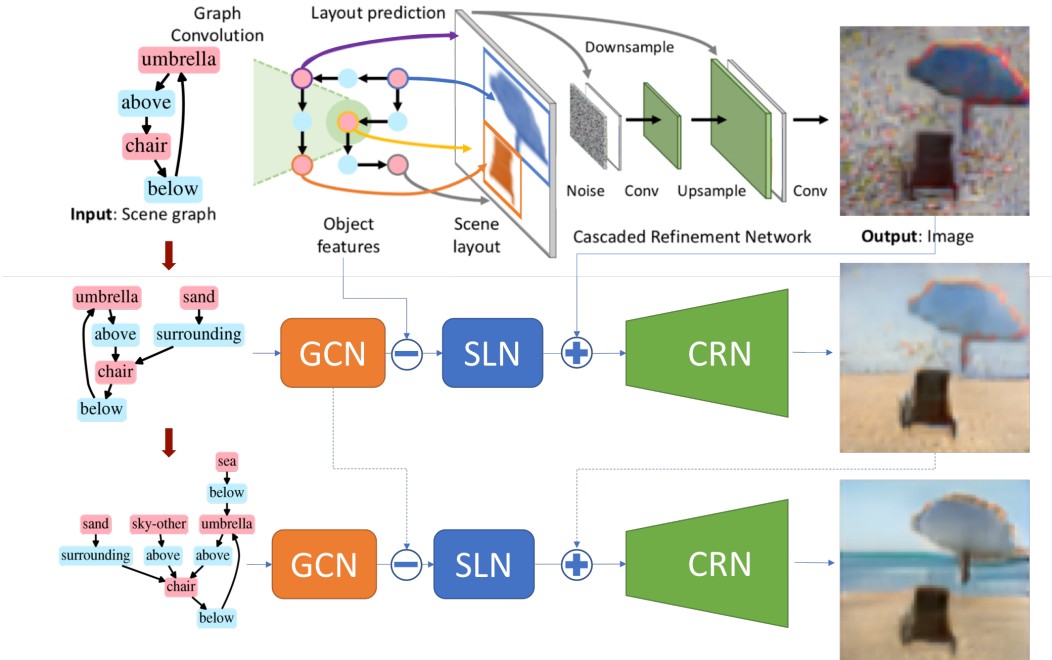

Figure 2: Model architecture for incremental image generation using scene graphs. The Graph Convolution Network takes as input the scene graph and produces object embeddings which are then fed to the Scene Layout Network. The SLN generates a layout by predicting the bounding boxes and segmentation masks for the objects. These are finally sent to the Cascaded Refinement Network which generates the image corresponding to the graph. This process is called iteratively to add objects in the image.

modules; which ensures that image generation is happening in a coarse-to-fine manner. The scene layout is first downsampled to the module input resolution and then fed to the module along with the previous module's output. Both these inputs are concatenated channel-wise and then passed to a pair of $3 \times 3$ convolution layers. This output is upsampled using nearest-neighbor interpolation and then passed to the next module. The output from the last module is finally passed to 2 convolution layers to produce the output image.

## 3.2 SEQUENTIAL GENERATION OF IMAGES WITH CONTEXT PRESERVATION

Our method allows for preserving context across the sequentially generated images by conditioning subsequent steps of image generation over certain information from previous steps.

- We extend Johnson et al. (2018) with a recurrent architecture that generates images using incrementally growing scene graphs using the components discussed in previous section as shown in Figure 2.
- To ensure that the image generated in the current step preserves the visual context from the previous steps, we replace three channels of the noise passed to CRN with the RGB channels of the image generated in the previous step. This encourages the CRN to generate the new image as similar as possible to the previously generated image.
- Moreover, we want the SLN to generate a layout corresponding to only the newly added objects in the scene graph. To this end, we remove the representations generated by GCN corresponding to the objects generated in previous steps.
- We do not have any ground truth for the intermediate generated images so we use perceptual loss for images generated in the intermediate steps to enforce the images to be perceptually similar to the ground truth final image. We do have L1 loss between the final image generated and the ground truth.

Concretely, we train the network with the following losses:

1. Adversarial losses: We use an image level and an object level discriminator to ensure realism of the images and presence of the objects. These are trained as in the regular GAN formulation : $\mathcal{L}_{GAN} = E_{x \sim p_{real}} log D(x) + E_{x \sim p_{fake}} log(1 - D(x))$

2. Box loss: Penalizes L1 distance between ground truth boxes from MS COCO vs the predicted labels as $\mathcal{L}_{box} = \sum_i^n ||b - b'||$

3. Mask loss: Penalizes difference between the masks predicted vs the ground truth masks, using cross entropy loss.

4. L1 pixel loss: Penalizes the difference between the ground truth image from MS COCO and the final generated image at the end of the incremental generation. L1 pixel losses are also used to penalize the difference between the previous and current generated image. $\mathcal{L}_{pixel} = ||I_G - I_{final}||$

5. Perceptual Similarity Loss: Serves as a regularization to ensure that the images generated at different steps are contextually similar to each other. Since we do not have ground truth for intermediate steps, we introduce an additional perceptual similarity loss using Zhang et al. (2018) between the final ground truth image and the different images generated in the intermediate steps. This allows the model to 'hallucinate' for the intermediate steps in a way that the contents of the image are similar to the ground truth. $\mathcal{L}_{perceptual} = ||P^\phi(I_k) - P^\phi(I_G)||^2$ where $P^\phi$ is the function computing a latent embedding that captures the perceptual characteristics of an image.

Thus we can provide additional supervision on the coordinates of the bounding boxes predicted by the layout network, to explicitly ensure the relations are preserved.

## 4 EXPERIMENTS

### 4.1 DATASET

We perform experiments on the 2017 COCO-Stuff dataset Caesar et al. (2016) which augments a subset of the COCO dataset (Lin et al., 2014) with additional stuff categories. The dataset annotates 40K train and 5K validation images with bounding boxes and segmentation masks for 80 'thing' categories (people, cars, etc.) and 91 'stuff' categories (sky, grass, etc.).

We follow the procedure described in Johnson et al. (2018) to construct synthetic scene graphs from these annotations based on the 2D image coordinates of the objects, using six mutually exclusive geometric relationships: left of, right of, above, below, inside, and surrounding. We create three splits for each image based on the number of objects in it. We randomly select 50% of the objects for the first split and incrementally add 25% objects for the next two splits. We then synthetically create separate scene graphs for each split. We train incremental generation for three steps, but this can be easily extended to more number of steps.

Note that previous works like El-Nouby et al. (2018) have relied on entirely synthetic datasets. For synthetic datasets, intermediate ground truth for all interim stages of the input text can easily be generated. However, for interactive image generation to be truly useful, it needs to generate realistic looking images not constrained to a synthetic dataset's subspace. The challenge with using real datasets like COCO is that we do not have "ground-truth" images corresponding to interim scene graphs (i.e., where all objects of the training image are not present). Only at the final step, with the full scene graph, do we get supervision from the training image. This makes our loss formulation described in the previous section particularly important.

To enable comparison against Johnson et al. (2018), we follow their dataset preprocessing steps. They ignore objects covering less than 2% of the image, and use images with 3 to 8 objects. They divide the COCO-Stuff 2017 validation set into their own validation and test sets, which contain 24,972 train, 1024 validation, and 2048 test images. For fair comparison, we do the same.

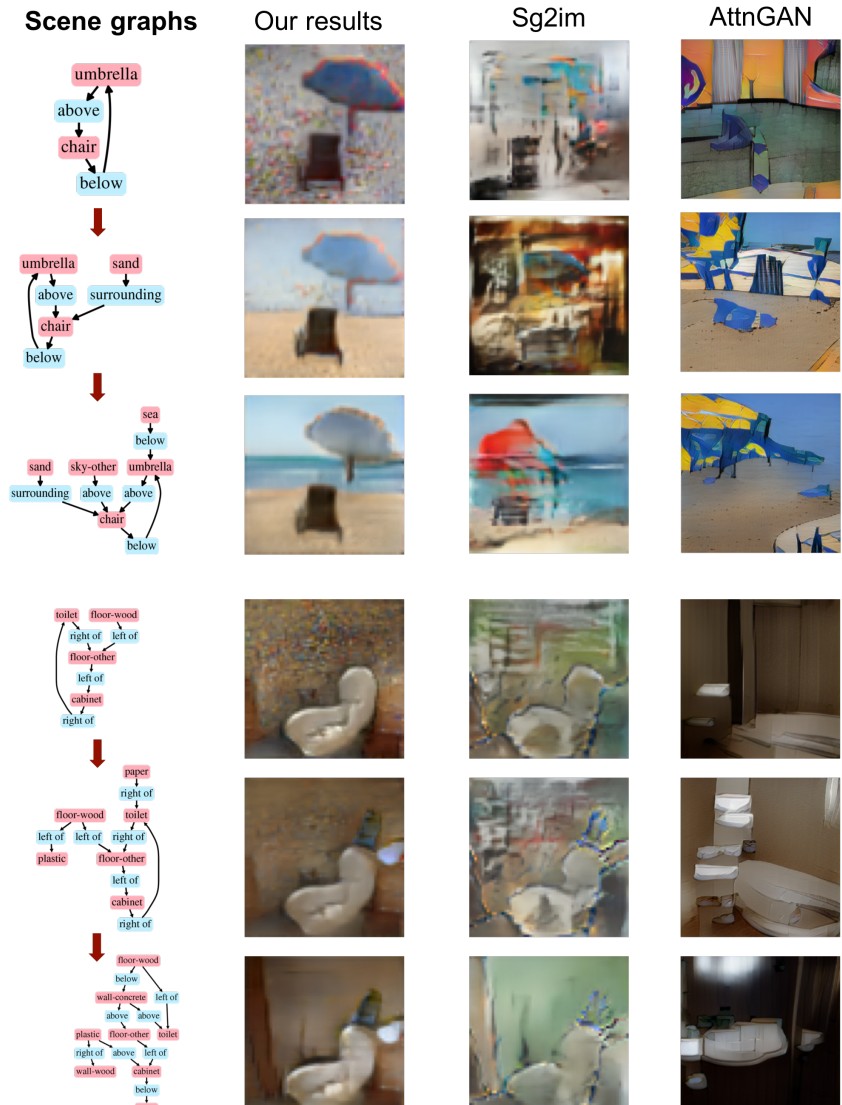

Figure 3: Comparison of our results with baseline approaches–sg2im and AttnGAN. While sg2im captures the scene graph's semantic context, it lacks consistency over multiple passes and the generated images are of poor quality. AttnGAN on the other hand generates higher resolution images but doesn't capture semantic context from the scene graph. Our model addresses both of these shortcomings by incrementally adding objects in accordance with the scene graph and generating higher quality images.

## 4.2 QUALITATIVE RESULTS

We compare the performance of our model against two baselines (Johnson et al., 2018) and (Xu et al., 2017) as can be seen in Figure 3. The scene graph in the first step contains two objects and the relationships between them. An additional object and its relationship with other objects is added to the scene graph at each of the next two steps. Note that the two baselines do not generate incrementally but for fair comparison, we generate outputs from them by feeding different amounts of information (scene graph or text) in three independent forward passes.

As can be seen, the first baseline Johnson et al. (2018) is able to capture the semantic context provided by the graph. However (i) it fails to preserve consistency over multiple passes and generates a completely new image for each scene graph, agnostic of what it had generated at the previous step and

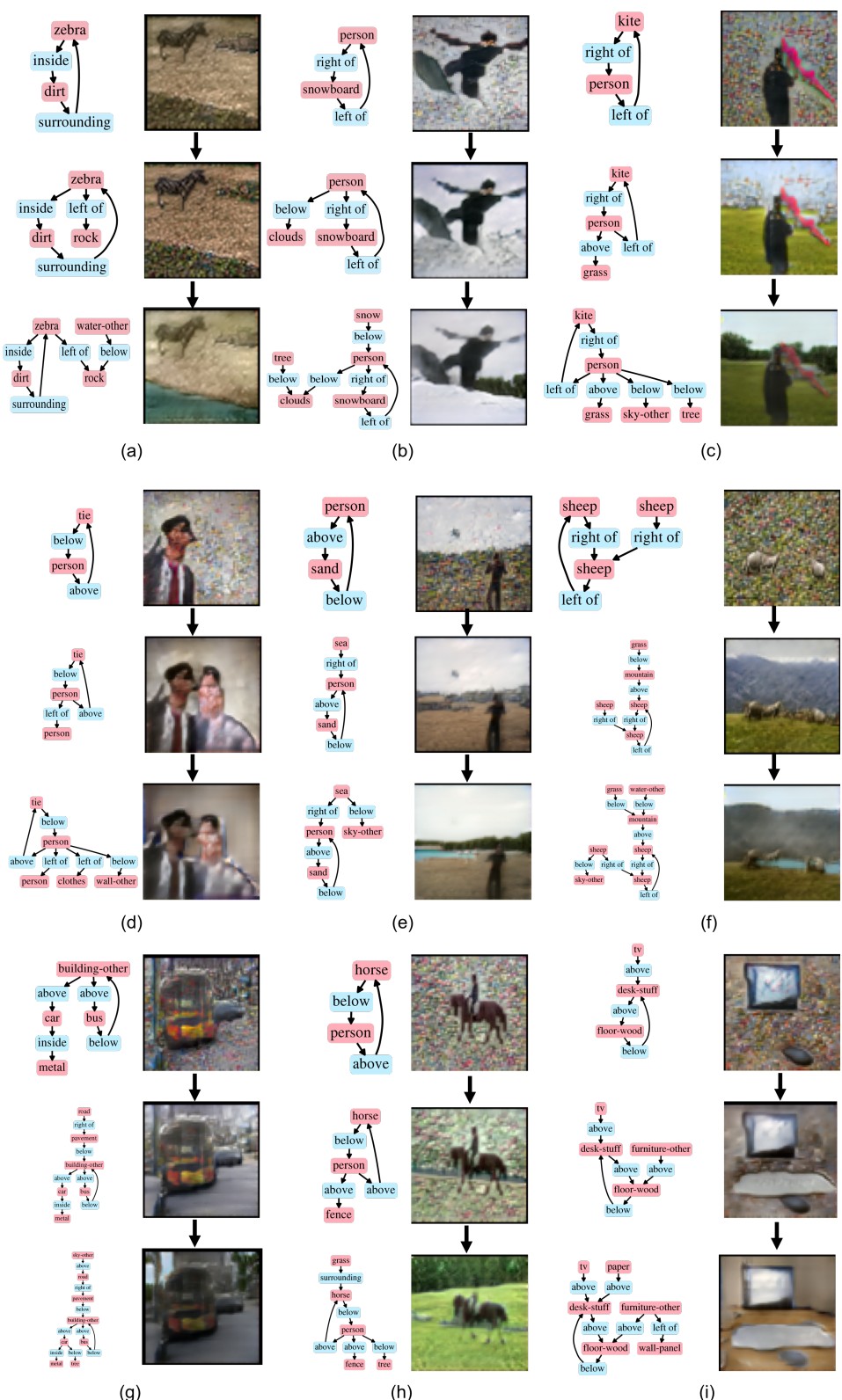

Figure 4: Sample outputs of our pipeline, visualized over 3 steps of generation. 3 splits for each image are created, based on the number of objects in that image. Synthetic scene graphs are then generated for each split and used for incremental image generation. While three steps are used for training, this method is quite easily extensible for more number of steps as well.

(ii) the images generate are of poor quality. The second baseline (Xu et al., 2017) produces visually pleasing and high resolution images but completely fails to capture any semantic context provided from the graph. Our model, on the other hand, is capable of incrementally adding new objects to the image created in the previous step in accordance with the relationships defined in the scene graph. Additionally, the quality of generated images is significantly improved, since at each step the model has to generate only a few objects rather than generating cluttered scenes with multiple objects, hence enabling it to better generate scene semantics.

We present comprehensive results generated by our model in Figure 4. It can be seen in Figures 4(c,f,h) that the model starts by generating objects as described by the scene graph and outputs noise in the background when it does not have enough information provided as input. As the background information is added in the scene graph, background objects like grass and sky begin to materialize. However it can also be seen that sometimes due to inherent biases in the dataset, the model begins to hallucinate the background even when no information is provided explicitly. In Figure 4(a), even though there is no mention of rock or water in the initial graph, the model is hallucinating objects of similar shapes at similar locations.

## 4.3 QUANTITATIVE EVALUATION

We use Inception Score (Salimans et al., 2016) for evaluating the quality of the images generated from our models. Inception Score uses an ImageNet based classifier to provide a quantitative evaluation of how realistic generated images appear. Inception Scores were originally proposed with the two main goal. Firstly, the images generated should contain clear objects (i.e. the images are sharp rather than blurry), (or, for image $x$ and label $y$, $p(y|x)$ should be low entropy). Secondly, the generative algorithm should output a high diversity of images from all the different classes in ImageNet, or $p(y)$ should be high entropy.

| Ground Truth | Johnson et al. (2018) | Step 1 (Ours) | Step 2 (Ours) | Step 3 (Ours) |
|---|---|---|---|---|
| 6.13 | 3.05 | **3.68** | **5.02** | **4.14** |

Table 1: Inception Scores for Ground Truth Images, images generated from Sg2im and the three steps from our model

The inception scores are reported in Table 1. We compare the inception score of the images generated from the baseline model sg2im with the full scene graph of the ground truth images. For our sequential generation model, we report the scores over three steps of generation, where at the third step the scene graph is the full scene graph corresponding to the ground truth image. We observe that due to our modified loss formulation and incremental generation, our model performs better in inception scores in all three steps.

Next, we evaluate how well our model retains the generated image from the previous step when adding in information to the image in the subsequent step (i.e., how "visually consistent" the generated images are). We use the perceptual similarity loss proposed by Zhang et al. (2018), and report the mean losses in Table 2. As seen both in the table and qualitatively in Figure 3, our model is much more successful at preserving context and previously generated content over subsequent steps.

| Method | Step 1 to Step 2 | Step 2 to Step 3 |
|---|---|---|
| Johnson et al. (2018) | 0.658 | 0.496 |
| Ours | **0.477** | **0.421** |

Table 2: Mean Perceptual similarity loss, evaluated between the images generated images of sg2im and our proposed model. Lower is better.

## 5 DISCUSSION AND FUTURE WORK

In this paper, we proposed an approach to sequentially generate images using incrementally growing scene graphs with context preservation. Through extensive evaluation and qualitative results, we

demonstrate that our approach is indeed able to generate an image sequence that is consistent over time and preserves the context in terms of objects generated in previous steps. In future, we plan to explore generating end-to-end with text description by augmenting our methodology with module to generate scene graphs from language input. While scene-graphs provide a very convenient modality to capture image semantics, we would like to explore ways to take natural sentences as inputs to modify the underlying scene graph. The current baseline method does single shot generation by passing the entire layout map through the Cascade Refinement Net for the final image generation. We plan to investigate whether the quality of generation can be improved by instead using attention on the GCN embeddings during generation. This could also potentially make the task of only modifying certain regions in the image easier. Further, we plan to explore better architectures for image generation through layouts for higher resolution image generation.

## ACKNOWLEDGEMENTS

We would like to thank Dr. Louis-Philippe Morency and Varun Bharadhwaj Lakshminarasimhan for their guidance and for providing us the opportunity to work on this project through the Advanced Multi-Modal Machine Learning (11-777) course from Language Technologies Institute (LTI) at Carnegie Mellon University.

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
