# OpenReview forum: "Interactive Image Generation Using Scene Graphs"
_ICLR.cc/2019/Workshop/DeepGenStruct — DeepGenStruct 2019_

### Official Review · AnonReviewer1 · 2019-04-11
**Good performance improvements over baselines, but not entirely clear what techniques are new**

**Rating:** 3
**Confidence:** 1

**Review:**

For scene-graph-to-image generation, this paper proposes some changes to the baseline (Johnson et al 2018) in order to produce the image iteratively. The scene-graph is accumulated over three steps, and for each step an image is produced. The intermediate images are not required; i.e. no intermediate supervision necessary. The proposed model conditions each step on the previous image, and introduces losses to encourage continuity in the sequence of images. The authors show performance improvements on COCO-Stuff in inception score (quality / realisticness) and mean perceptual similarity loss (consistency across steps) compared to Johnson et al.

Pros:
- Performance improvements seem solid, and the examples in Figures 3 and 4 seem convincing.
- It seems that this work has some novelty in that it is the first to generate real-world images iteratively without intermediate supervision

Cons:
- The description of the baseline and proposed model (sections 3.1 and 3.2) would benefit from some more mathematical detail, i.e. introduce some notation in 3.1 so that in 3.2 you can precisely explain how you are changing the model.
- As a person who is not familiar with the related work, I was unsure what techniques are new here. The bullet-point list in section 3.2 seems to be a list of the differences between the baseline and the proposed model. As far as I can tell, these differences seem quite minor, except for the first bullet point which sounds like it might be quite complicated (but it's not described in any detail, so perhaps it is simple). This is where some more precise mathematical notation would be useful. Similarly, in the list of losses in section 3.2 I'm not sure what's part of the baseline and what's new.
- In particular, while reading the paper I was under some uncertainty about whether the baseline methods (Johnson et al 2018 and Xu et al 2017) are "iterative". Some lines imply they are iterative in some sense ("AttnGANs also begin with a low-resolution image, and then improve it over multiple steps to come up with a final image" / "The layout is passed to a cascaded refinement network which generates an output image at increasing spatial scales."), but this paper claims to be the first to iteratively generate real-world images without intermediate supervision. So I'm unsure what exactly is different about this paper compared to previous.
- This paper doesn't include any human evaluation - instead relying on automatic metrics only. For comparison, Johnson et al 2018 include some human evaluation.

Other comments:
- "We also note that Stage 1 performs the best. From our observations this is because the vividness of the image colors and object definitions is the best at the stage 1, and begin to fade out stage 2 onwards." Though this explains some more fine-grained ways in which stage 1 is the best (vividness, object definitions), it doesn't explain *why* stage 1 is the best (i.e. why is the model most able to make realistic images on the middle step? If it can make realistic images on step 1 why can't it improve on them on step 2?)
- "coarse-to-fine" not "course-to-fine"

Note: Though I am familiar with Deep Learning, I am not very familiar with computer vision, so it is possible that I am missing something in my reading of this paper.

---

### Official Review · AnonReviewer2 · 2019-04-15
**interactive image generation using scene graphs**

**Rating:** 3
**Confidence:** 2

**Review:**

This paper proposes a conditional adversarial model that iteratively generates images given a scene graph. The scene graph describes the relations between the different objects and components of the image. It is shown that images can be generated iteratively by augmenting the scene graph with new objects and relations, and the existing image content will be maintained.

This work uses a combination of many different building blocks that have recently gained traction in literature, including graph convolutional networks to process the scene graphs, networks for bounding box prediction and conditional generative adversarial networks. These are combined with a variety of loss functions (5 in total).

The resulting system is shown to work reasonably well, but it is quite complex and I feel that the importance of each individual component could be demonstrated better by including some ablations -- what would happen if a GAN were conditioned directly on the output of the GCN that processes the scene graph, for example? Is the intermediate step that produces segmentations and bounding boxes strictly necessary?

In two different places in the manuscript, it is stated that the model is the first of its kind to the authors' knowledge. I find such statements a bit inappropriate when they refer to very specific problem settings. There has definitely been a lot of closely related work e.g. on image generation conditioned on captions (including some that uses scene graphs as an intermediate representation). Stating that the work is presumably the first to use this particular specific combination of input representations and model structure is not very meaningful.

The manuscript contains quite a few grammatical and spelling errors and would benefit from proofreading.

---

### Decision · Program_Chairs · 2019-04-19
**Acceptance Decision**

**Decision:**

Accept

**Comment:**

Accepted